# MASleepNet: A Sleep Staging Model Integrating Multi-Scale Convolution and Attention Mechanisms

**DOI:** 10.3390/biomimetics10100642

**Published:** 2025-09-23

**Authors:** Zhiyuan Wang, Zian Gong, Tengjie Wang, Qi Dong, Zhentao Huang, Shanwen Zhang, Yahong Ma

**Affiliations:** 1Xi’an Key Laboratory of High Precision Industrial Intelligent Vision Measurement Technology, School of Electronic Information, Xijing University, Xi’an 710123, China; 15716560035@163.com (Z.W.); 15906232055@163.com (Z.G.); 15847613111@163.com (T.W.); huangzhentao168@163.com (Z.H.); 2School of Mathematics and Statistics, Zhengzhou University, Zhengzhou 450001, China; qid004841@gmail.com; 3School of Electronic Information, Xijing University, Xi’an 710123, China

**Keywords:** multi-channel PSG, deep learning, BiLSTM, attention mechanism

## Abstract

With the rapid development of modern industry, people’s living pressures are gradually increasing, and an increasing number of individuals are affected by sleep disorders such as insomnia, hypersomnia, and sleep apnea syndrome. Many cardiovascular and psychiatric diseases are also closely related to sleep. Therefore, the early detection, accurate diagnosis, and treatment of sleep disorders an urgent research priority. Traditional manual sleep staging methods have many problems, such as being time-consuming and cumbersome, relying on expert experience, or being subjective. To address these issues, researchers have proposed multiple algorithmic strategies for sleep staging automation based on deep learning in recent years. This paper studies MASleepNet, a sleep staging neural network model that integrates multimodal deep features. This model takes multi-channel Polysomnography (PSG) signals (including EEG (Fpz-Cz, Pz-Oz), EOG, and EMG) as input and employs a multi-scale convolutional module to extract features at different time scales in parallel. It then adaptively weights and fuses the features from each modality using a channel-wise attention mechanism. The integrated temporal features are integrated into a Bidirectional Long Short-Term Memory (BiLSTM) sequence encoder, where an attention mechanism is introduced to identify key temporal segments. The final classification result is produced by the fully connected layer. The proposed model was experimentally evaluated on the Sleep-EDF dataset (consisting of two subsets, Sleep-EDF-78 and Sleep-EDF-20), achieving classification accuracies of 82.56% and 84.53% on the two subsets, respectively. These results demonstrate that deep models that integrate multimodal signals and an attention mechanism offer the possibility to enhance the efficiency of automatic sleep staging compared to cutting-edge methods.

## 1. Introduction

Sleep is a vital physiological process, occupying over one-third of the human lifespan. Good sleep plays a vital role in maintaining both physical and mental health [1]. However, in modern society, factors such as life stress, excessive use of electronic devices, and circadian rhythm disorders have led to the increasing prevalence of insomnia, sleep apnea, and other problems, seriously endangering people’s health and quality of life [2,3]. With the increasing attention paid to sleep health issues, the study of sleep staging has gradually become a major research focus in sleep medicine [4,5,6]. Traditional sleep staging studies require professional physicians to manually segment the polysomnography (PSG) signal and divide it into five stages (W, REM, N1, N2, N3) as defined by internationally accepted sleep staging standards [7]. This method is highly time-consuming, requires substantial human effort, and often suffers from inconsistencies across different scorers [8,9,10]. Therefore, developing an automated sleep staging algorithm holds substantial significance.

Significant advancements in classification have been driven by the integration of machine learning and deep learning methods in recent years [11,12,13,14,15,16,17]. However, most early studies mainly used single-channel electroencephalogram (EEG) signals [18,19] and relied on expert prior experience for feature extraction. A single EEG channel is insufficient to comprehensively capture the diverse physiological changes during sleep. For example, changes in eye movement and muscle tension are also crucial for identifying stages such as REM. Therefore, methods that integrate multimodal signals are expected to increase the efficiency of automatic sleep staging. Deep neural network models have proven effective in sleep staging tasks [20,21,22]. For example, DeepSleepNet, proposed by Supratak et al. [23], integrates CNNs with Bidirectional Long Short-Term Memory (BiLSTM) networks to automatically extract features and model stage sequences from single-channel EEG raw data. The model extracts features of different frequency bands such as slow waves and fast waves through multi-scale convolution, and improves staging performance by combining sequence information.

A series of subsequent works further improved on this basis. For example, Eldele et al. [24] proposed the AttnSleep model, which introduced multi-resolution convolutional feature extraction mechanism to capture features of different frequencies and adjust feature channel weights, thereby improving the quality of feature representation. At the same time, by modeling long-range dependencies through the attention mechanism, the accuracy of single-channel staging and the ability to detect each stage were effectively improved. The 1D-ResNet-SE-LSTM model proposed by Li et al. [25] combines the residual network with the SE attention module, works with LSTM to capture stage transition patterns, and introduces a weighted loss function to reduce the impact of category imbalance. The above studies show that multi-scale convolution, channel attention, and temporal context modeling are crucial to improving sleep staging performance.

In terms of multi-channel and multi-modal signals, some automatic staging models that integrate EEG, EOG, and EMG have also emerged in recent years. For example, the SleepPrintNet proposed by Jia et al. [26] uses a multivariate and multimodal neural network to capture salient features from multiple physiological modalities, significantly improving the staging accuracy. Subsequently, they proposed the SalientSleepNet, which further achieved a parameter-efficient and performance-leading staging model through multimodal saliency wave detection and multi-scale time series feature extraction [27]. The above work proves the effectiveness of multimodal data fusion and attention mechanism in sleep staging.

Deep learning has made remarkable advances in automatic sleep staging, but the following major challenges still exist: (1) Single-channel EEG-based models have difficulty fully utilizing complementary information from modalities; (2) An uneven distribution of sleep data among different stages is observed, resulting in insufficient recognition performance of the model for minority classes, especially the N1 stage [28,29]; (3) The cross-scale fusion of time-frequency features and the modeling of long-range temporal dependencies are still insufficient, which makes it difficult for the model to learn complex sleep dynamics.

To overcome these limitations, this study introduces a MASleepNet sleep staging model. This model integrates multi-scale convolution, channel attention, temporal Bidirectional Long Short-Term Memory (BiLSTM), an attention mechanism, and a class weighting strategy into a unified architecture to automatically classify sleep stages from multi-channel polysomnography (PSG) signals. Compared to existing methods, this model strives to improve overall accuracy while strengthening the distinguish of the minority class N1.

The main contributions of the study are: (1) Multimodal and multiscale feature extraction: The dual-branch convolutional network is designed to process different time domain scales in parallel, extracting cross-band features for EEG/EOG/EMG signals while taking into account transient spikes and slow wave patterns. One branch uses a larger convolution kernel to capture slow wave information spanning several seconds, while the other branch uses a smaller one to capture sub-second transient features. The outputs of the two branches are spliced to obtain a rich time-frequency feature representation. (2) Channel attention fusion: The Squeeze-and-Excitation (SE) module is applied to each modal convolution feature to achieve channel-level adaptive weighting. The SE module learns channel importance weights by applying global pooling and subsequent fully connected network to highlight key channel features. This mechanism can effectively enhance discriminative channel information such as REM stage eye movement features and N3 stage slow wave features. (3) BiLSTM+attention temporal sequence modeling: The fused multimodal feature sequence is input into a BiLSTM to capture temporal correlations, and then the LSTM output is weighted and summed through the attention mechanism to screen the time segments that contribute most to the current epoch judgment. The BiLSTM’s forward and backward hidden states can simultaneously utilize contextual information, helping to learn the evolution of sleep stages. The attention layer assigns weights to features at different moments in accordance with the trained query vector, allowing the model to automatically ignore irrelevant information and highlight key features.

The organization of this paper is as follows: Section 2 introduces the MASleepNet model architecture and its key components. Section 3 presents the dataset, experimental setup, evaluation metrics, and results. Section 4 discusses ablation studies. Section 5 provides the conclusions, and Section 6 outlines future work.

## 2. Methods

This section describes the MASleepNet model for classification using multimodal data, including its overall framework and key components. The model uses multi-channel polysomnography (PSG) signals as input.

### 2.1. Overall Model Architecture

The architecture of the proposed MASleepNet sleep staging model is illustrated in Figure 1. It has three functional modules: (1) Multi-modal Scalable Feature Extraction Module (MFEM); (2) Bi-directional Time Series Modeling and Focusing Module (BiLSTM-Attention Context Module, ACM); and (3) Sleep Stage Classifier (Classifier).

MASleepNet receives polysomnography (PSG) signals comprising four channels in total: two EEG channels (Fpz–Cz, Pz–Oz), one EOG channel, and one EMG channel. It first extracts multimodal time-frequency features using a multi-scale convolutional network. A channel-wise attention mechanism (SELayer) is then introduced after each modal channel for weighted fusion. The integrated multimodal features are subsequently input into a bidirectional LSTM + Attention module (BiLSTM-Attention) to extract contextual information and focus on temporal features at key moments. Finally, the classification results are output through the fully connected layer.

The entire model architecture is end-to-end trainable. The components of the model are designed to work synergistically: the MFEM extracts modal features across time scales, the SE channel attention highlights key channels, and the temporal evolution and focuses on discriminative key segments, thereby improving overall classification performance.

### 2.2. Multimodal Multiscale Feature Extraction Module (MFEM)

The MFEM is designed to extract modality-specific features from EEG, EOG, and EMG signals. As shown in Figure 2. This module constructs independent convolutional feature extraction branches for the feature, EOG, and EMG signals. Each modality branch uses a parallel dual-branch one-dimensional convolutional network to capture key information at multiple time scales, as shown in Figure 2. Multi-branch structures have been widely shown in previous studies to enhance model robustness and generalization [30]. Although multi-branch structures have certain advantages in feature extraction, introducing too many branches will significantly increase computational costs and may even lead to side effects such as overfitting. Existing studies have shown that additional branches have limited impact on performance improvement [31]. Based on this consideration, this paper adopts two convolutional branches in the model design to balance classification accuracy and computational efficiency.

Specifically, the branch of each modality contains the following two convolution branches:(1)Short-time-scale convolution branch (local feature branch): This path uses a smaller convolution kernel (kernel size = 50) to accurately capture short-time local information in the signal, such as the wake phase (W) and high-frequency components like spindles;(2)Long-time convolution branch (global feature branch): This path uses a larger convolution kernel (kernel size = 400) and the receptive field covers a signal length of about 4 s to fully extract the slower trend features in the signal, such as the slow wave features in the N3 stage.

The basic structure of each convolution branch includes convolution, batch normalization (BN), LeakyReLU activation, max pooling, and dropout regularization. The specific convolution calculation process can be defined as Formula (1):(1)$$X_{j}^{l+1}=f\left(\sum_{i\in M_{j}}\;X_{i}^{l}\times W_{ij}+b_{j}\right)$$

Among them, $X_{i}^{l}$ represents the i channel of the l layer input feature, $W_{ij}$ denotes the weight and $b_{j}$ corresponds the bias of the convolution kernel, and f(⋅) is the LeakyReLU activation function.

After the two convolution branches extract features respectively, they are concatenated in the time dimension to form a fused multi-scale feature.

Furthermore, considering that EEG, EOG, and EMG signals have different levels of importance in identifying different stages (for example, EOG eye movement features are more critical in the REM stage), this study introduced the a SE channel attention module following the convolutional fusion features of each modality to achieve adaptive weighting of channel features and reduce the inevitable redundancy generated when extracting large-scale features and small-scale features [32]. The SE module first captures the statistical characteristics of each channel via global average pooling. The calculation process is shown in Formula (2):(2)$$z=\frac{1}{T}\sum_{t=1}^{T}\;X[:,t]$$

Then, the attention weight vector s of each channel is generated using two successive fully connected layers. The specific calculation formula is (3):(3)$$s=\sigma \left(W_{2}\delta \left(W_{1}z\right)\right)$$

Finally, the weight vector is multiplied element-by-element with the original channel feature to achieve adaptive channel weighting of the feature, so as to highlight the channel features with stronger discriminative ability and suppress the interference of redundant or noisy channels. The final channel-weighted feature is expressed as Formula (4):(4)$${\hat{X}}_{c}=X_{c}\cdot s_{c},c=1,2,\ldots ,C$$

Among them, $W_{1}$ and $W_{2}$ are the learnable weight parameter matrices, σ denotes the sigmoid activation function, and δ corresponds to the ReLU activation function.

Through the above feature extraction process, the weighted features of EEG (2 channels), EOG (1 channel), and EMG (1 channel) are concatenated along the channel dimension to construct a unified multimodal fused feature tensor. This fused feature tensor is further rearranged into a format suitable for sequence modeling (i.e., (batch size, time steps, features)) to be passed to the subsequent bidirectional LSTM and attention focusing modules for sequential modeling and classification of sleep stages.

### 2.3. BiLSTM-Attention Context Module (ACM)

Sleep stage sequences exhibit strong contextual relevance in the temporal dimension, with strong dependencies across different stages. To effectively model this long-range temporal dependency, this study designed a BiLSTM-Attention Context Module (ACM). This module combines a BiLSTM network with an attention mechanism to enhance context modeling capabilities in sequential data. Such architectures have been extensively applied in diverse domains.

The first employs a Bi-LSTM network to simultaneously capture contextual information of the feature sequence in both forward and backward directions. As illustrated in Figure 3, the input feature sequence X = (x_1,x_2,…,x_T), obtained after multi-modal feature fusion, is fed into two LSTM layers: one processes the sequence in the forward direction while the other in the backward direction. At each time step t, the hidden states from both directions are concatenated to form the bidirectional representation, as expressed in Formula (5):(5)$$h_{t}=\left[\vec{h_{t}};\overset{\leftarrow }{h_{t}}\right],t=1,2,\ldots ,T$$

Subsequently, a sequence-wise attention module is employed to automatically focus on the most discriminative key segments in the temporal feature sequence. The specific steps involved in the calculation of the attention mechanism are outlined below.

First, the attention weight $\alpha_{t}$ of the hidden state $h_{t}$ at each time step as shown in Formula (6):(6)$$\alpha_{t}=\frac{\exp\left(v^{⊤}tanh\left(W_{h}h_{t}+b_{h}\right)\right)}{\sum_{t^{\prime }=1}^{T}\;exp\left(v^{⊤}tanh\left(W_{h}h_{t^{\prime }}+b_{h}\right)\right)}$$

Among them, the attention mechanism employs $W_{h}$ as the weight matrix and $b_{h}$ as the associated bias term, v denotes the attention vector parameter, tanh(⋅) is the hyperbolic tangent activation function, and exp(⋅) represents the exponential function.

Subsequently, according to the above weights, the hidden state sequence output by BiLSTM is weighted and summed to form the final attention context vector c, as shown in Formula (7):(7)$$c=\sum_{t=1}^{T}\;\alpha_{t}h_{t}$$

Through these calculations, the automatically prioritizes the time segments that contribute most to classification, while suppressing less contributive or redundant features. This module demonstrates a significant performance advantage in the identification of easily confused sleep stages, including N1 and REM.

### 2.4. Classifier and Loss Function

The classifier of the model maps the attention context feature vector c output by the to the final classification space.

To avoid model overfitting, a Dropout regularization strategy is used on the context vector c, with a Dropout ratio of *p* = 0.5 set to obtain the regularized feature vector representation $c^{\prime }$. The regularized context feature vector is input to the fully connected classification layer, and the predicted class probabilities of the model distribution for each sleep stage is finally obtained y. The output calculation process of the classifier can be expressed as Formula (8):(8)$$y=softmax\left(W_{fc}c^{\prime }+b_{fc}\right)$$

Among them, the weight parameter matrix is represented by $W_{fc}$, the bias vector of the fully connected layer is indicated by $b_{fc}$, and the softmax(⋅) function is used to convert the linear output into a probability form to denote the model’s predicted probabilities for each sleep stage class.

The cross-entropy loss quantifies the divergence between the predicted distribution y and the actual labels $\hat{y}$. The definition of the loss function is provided in Formula (9):(9)$$L\left(\hat{y},y\right)=-\sum_{k=1}^{K}\;{\hat{y}}_{k}log\left(y_{k}\right)$$
where K = 5 represents the complete set of sleep stage categories, ${\hat{y}}_{k}$ is the one-hot encoding vector of the true label, and $y_{k}$ is the probability value predicted by the classifier.

## 3. Experiments and Results

### 3.1. Dataset and Preprocessing

This study utilized the Sleep-EDF dataset provided by PhysioNet [33,34], which comprises two subsets, Sleep-EDF-20 and Sleep-EDF-78 (Table 1). Sleep-EDF-20 includes two-night recordings from 20 subjects, Sleep-EDF-78 is an extended dataset containing single-night recordings from 78 subjects. Each polysomnography (PSG) recording comprises two EEG channels (Fpz-Cz, Pz-Oz), one EOG channel, and one submental (chin) EMG channel, all sampled at 100 Hz. The original dataset was annotated by experts following the AASM guidelines into five stages: W, N1, N2, N3, and REM [35]. During preprocessing, each recording was divided into epochs and annotated accordingly. For Sleep-EDF-20 and Sleep-EDF-78, this study the same preprocessing pipeline was applied and without extracting additional intervals.

### 3.2. Experimental Plan

To evaluate the generalization ability of MASleepNet, this study employed 5-fold cross-subject validation. The data from 20 subjects were randomly partitioned into five subsets, with four subsets used for training and validation in each iteration, and the remaining subset reserved as test set. To mitigate overfitting and enable early stopping, 10% of the training samples were randomly held out as a validation set. During training, the validation loss was continuously monitored. Training was terminated if the validation loss did not decrease for five successive epochs, and the model parameters corresponding to the best result were saved.

Model training was conducted in an end-to-end manner. The Adam optimizer was selected due to its widespread application in sleep staging and its stable convergence properties. The initial learning rate was set to 0.001, which is commonly adopted in related studies and was further confirmed through preliminary tuning experiments to provide a balance between convergence stability and computational efficiency. The cross-entropy loss function was employed since it is well-suited for multi-class classification tasks. A batch size of 64 was chosen to ensure both computational efficiency and gradient stability, while the maximum number of epochs was limited to 100, which was sufficient for convergence under the given settings. These hyperparameter configurations were determined based on prior research and empirical validation, ensuring consistent and reliable training performance.

### 3.3. Performance Evaluation Index

The evaluation metrics of the model are: overall accuracy (ACC), F1 score (F1_Macro_), and Kappa coefficient. Sensitivity (SEN), specificity (SPE), and precision (PRE) were also calculated.

The accuracy score is calculated using Formula (10):(10)$$ACC=\frac{TP+TN}{TP+FP+TN+FN}$$

The F1 is calculated using Formula (11):(11)$${F1}_{Macro}=\frac{1}{K}\sum_{k=1}^{K}\;\frac{2\cdot PRE_{i}\cdot SEN_{i}}{PRE_{i}\cdot SEN_{i}}$$

The Kappa coefficient is calculated using Formula (12):(12)$$Kappa=\frac{P_{0}-P_{e}}{1-P_{e}}$$

Here, $P_{0}$ represents observed consistency and $P_{e}$ represents random expected consistency.

Sensitivity and specificity are calculated according to Formula (13):(13)$$SEN=\frac{TP}{TP+FN},\;SPE=\frac{TN}{TN+FP}$$

The precision is calculated using Formula (14):(14)$$PRE=\frac{TP}{TP+FP}$$

In the formula above, TP denotes true positives, TN denotes true negatives, FP denotes false positives, and FN denotes false negatives.

### 3.4. Classification Performance of the MASleepNet Model

To intuitively demonstrate the model’s prediction performance at the individual subject level, we visualized the complete sleep process of subject SC406 from the SleepEDF-78 dataset, as shown in Figure 4. As illustrated in Figure 4, panel (a) depicts the subject’s truth hypnogram, and panel (b) illustrates the hypnogram predicted by the proposed model.

Figure 4 clearly shows that the model’s output hypnograms closely match the ground-truth annotations in terms of overall trends and cyclical variations, demonstrating particularly good fit for the identification of N3, REM, and W stage. While some localized errors exist in individual stages, such as N1, the overall trend remains highly consistent, demonstrating the model’s robust performance in staging complex individual sleep structures.

Figure 5 and Figure 6 present the overall confusion matrices of MASleepNet on the Sleep-EDF-78 and Sleep-EDF-20 datasets under five-fold cross-validation. The results show that stages W, N2, N3, and REM achieve consistently high classification accuracy in both datasets, confirming the model’s strong ability to recognize the major sleep stages. In contrast, the N1 stage remains the most challenging, as reflected by the higher proportion of off-diagonal elements in its row. This indicates frequent misclassification of N1 as the adjacent W, N2, or REM stages, which can be attributed to its short duration and smooth signal transitions that overlap with neighboring stages.

Despite differences in sample size, signal composition, and subject diversity between the two datasets, MASleepNet maintains stable performance across both, demonstrating strong cross-dataset generalization. The overall distribution patterns of the confusion matrices remain consistent, while the slightly higher N1 confusion observed on Sleep-EDF-78 may be due to greater inter-subject variability. These results confirm that MASleepNet achieves competitive accuracy while preserving robustness and adaptability across heterogeneous datasets.

### 3.5. Baseline Networks and Comparison

As a reference, the study incorporated a classical and widely recognized sleep staging model for comparative evaluation such as AttnSleep [24], DeepSleepNet [23], ResNetLSTM [36], SeqSleepNet [15] and TinySleepNet [37] as comparison methods, namely: (1) AttnSleep adopts multi-resolution convolution and sequence attention mechanism, and is good at capturing long-range features; (2) DeepSleepNet: combines CNN and BiLSTM, and introduces end-to-end deep structure for the first time; (3) ResNetLSTM: uses ResNet structure to extract local features and combines LSTM to capture temporal information; (4) SeqSleepNet: a two-layer BiLSTM structure, and introduces attention mechanism at the frame level and stage level; (5) TinySleepNet: a lightweight CNN network that takes into account both computational efficiency and classification accuracy.

All comparison models were tested under the same data partitioning, experimental configuration, training parameters, and adopted the same experimental and evaluation settings as this study to ensure fairness and comparability of the results. Table 2 and Table 3 summarize the overall classification performance of MASleepNet and other baseline models on the Sleep-EDF-20 and Sleep-EDF-78 datasets, respectively, while Table 2 and Table 3 present the per-class precision results for each sleep stage.

An examination of Table 2 and Table 3 reveals that the proposed MASleepNet consistently achieves excellent performance, including accuracy, F1_Macro_, Kappa, sensitivity, specificity, and precision. In particular, the MASleepNet model significantly outperforms other methods in challenging sleep stages such as N1 and REM, demonstrating its effectiveness in capturing subtle and discriminative multimodal temporal features.

Specifically, MASleepNet achieved an average accuracy of 84% and an F1_Macro_ of 0.78 on the Sleep-EDF-20 test set; and an average accuracy of 82% and an F1_Macro_ of 0.76 on the Sleep-EDF-78 test set. Compared to methods such as AttnSleep, which only use single-channel EEG, this model improves both accuracy and F1_Macro_ (AttnSleep achieves approximately 81% accuracy on Sleep-EDF-20, but only approximately 0.37 F1_Macro_ for the minority class N1). In particular, MASleepNet maintains high accuracy for the major phases (N1, N3, and REM), with N3 performing best with an accuracy exceeding 85%. Misclassifications primarily occur between N3 and N2, and between REM and N1 (these phases share similar transition signals). The sensitivity of MASleepNet in recognizing the most challenging N1 stage is significantly greater than that of competing models. On the Sleep-EDF-20 dataset, MASleepNet’s accuracy for the N1 stage increased to 53.55%, significantly outperforming other models. This demonstrates that the incorporation of EOG and EMG information, along with the attention mechanism, effectively alleviates the difficulty in identifying the N1 stage. This model achieved classification accuracy exceeding 80% for all stages except N1 (nearly 90% for wake, N2, and N3). While approximately 40% of N1 samples were misclassified as N2 or REM, the overall false positive rate was lower than that of the baseline model. Comprehensive metrics and comparisons confirm that MASleepNet achieves superior classification capability and balanced performance in the automatic sleep staging task. This experiment validates MASleepNet as an effective and robust method for automatic classification, outperforming existing current leading approaches and providing data support for clinical applications.

## 4. Ablation Experiment

To gain deeper insights into the contribution of individual components, this study designed the following ablation experiments:(1)No-MSConv (no multi-scale convolution structure): This variant removes the large kernel convolution branch in the multi-scale feature extraction module and only retains the small kernel (local) convolution path in the original network to evaluate the impact of different receptive field combinations on feature extraction capabilities.(2)EEG-Only: This variant starts from the input layer and only retains EEG as input, completely removing the EOG and EMG signal paths, and correspondingly deleting the feature extraction and attention fusion modules of these two modalities. It is used to evaluate the gain effect of multimodal signal fusion on classification performance.(3)No-SE (remove squeeze-and-excitation module): This variant removes the SE channel attention module in each branch, such that feature maps are directly propagated without channel-wise recalibration. This design evaluates the effect of adaptive channel weighting on performance.(4)No-LSTM (remove BiLSTM): This variant removes the BiLSTM temporal modeling module and instead directly applies attention-based global pooling over the extracted features. It is used to evaluate the contribution of recurrent temporal modeling to sequence classification.(5)No-Attn (remove attention mechanism): This variant retains the BiLSTM module for time series modeling, but removes the attention mechanism; the classifier only receives the last time step in the LSTM output sequence as the global representation vector for classification, which is used to compare the attention context modeling ability.

Table 4 and Table 5 present the ablation results of the MASleepNet model and its variants on the Sleep-EDF-20 and Sleep-EDF-78 datasets, with detailed comparisons across Accuracy, F1Macro, Kappa, SEN, SPE, and PRE.

Comparing the results of EEG-Only and MASleepNet, we found that the introduction of multimodal fusion significantly improved classification performance, increasing accuracy by 1.80% and 2.62% on the Sleep-EDF-20 and Sleep-EDF-78 datasets, respectively. This strongly supports that EOG and EMG provide effective information complementary to EEG, playing a particularly positive role in the identification of easily confused phases such as N1 and REM.

Comparing the EEG-Only and MASleepNet models, multimodal fusion improved accuracy by 1.80% and 2.62% on Sleep-EDF-20 and Sleep-EDF-78, respectively, confirming the complementary role of EOG and EMG in supporting EEG, particularly in the recognition of challenging stages such as N1 and REM.

Removing the SE module (No-SE) led to a decline in overall performance (ACC decreased by 0.31% and 0.51%), demonstrating that adaptive channel weighting improves the discriminability of multimodal features by highlighting informative channels.

Eliminating the BiLSTM (No-LSTM) resulted in accuracy reductions of 0.35% and 1.04% on the two datasets, highlighting the importance of recurrent temporal modeling for capturing sequential dependencies in sleep signals.

Finally, the No-Attn variant also showed decreased accuracy (−0.40% and −0.71%), indicating that the attention mechanism further enhances feature utilization by focusing on discriminative time segments, especially improving recognition of the minority N1 stage.

Through comparative analysis, the effectiveness and necessity of the five key modules—multi-scale convolution, multimodal fusion, SE attention, BiLSTM, and attention mechanism—are validated, and their specific contributions to the overall performance of MASleepNet are systematically demonstrated.

## 5. Conclusions

This paper addresses the shortcomings of existing sleep staging models in modality fusion, multi-scale feature extraction, and sequential information capture by proposing a novel multimodal deep learning framework, MASleepNet. The model integrates a multi-scale convolutional neural network, a channel-level SE attention mechanism, and a BiLSTM network with a sequence-level attention mechanism. By leveraging the complementary characteristics of EEG, EOG, and EMG signals, MASleepNet effectively captures both local and global features across different time scales, as well as the long-term sequential dependencies between sleep stages.

Experimental results demonstrate that MASleepNet delivers superior performance on both the Sleep-EDF-20 and Sleep-EDF-78 datasets, significantly outperforming widely recognized baseline models such as AttnSleep, DeepSleepNet, ResNetLSTM, SeqSleepNet, and TinySleepNet. The model achieves notable improvements in overall accuracy, F1 score, and Cohen’s kappa, and shows remarkable effectiveness in recognizing the challenging N1 and REM stages. Ablation studies further confirm the contributions of each module: the multi-scale convolutional structure captures frequency-specific sleep characteristics, the SE attention module enhances discriminative feature channels, and the sequence-level attention mechanism identifies critical temporal dependencies. Collectively, these findings substantiate the effectiveness of the MASleepNet framework and highlight its potential for advancing sleep stage classification.

## 6. Future Work

Future research will further emphasize improving the computational efficiency and lightweight design of the model, with the aim of enabling real-time deployment in portable and embedded sleep monitoring devices. Additionally, validating the model’s generalizability on diverse and large-scale clinical datasets will be a key focus, ensuring its applicability to real-world medical environments. Extending the framework to integrate additional physiological signals and exploring advanced fusion strategies may also enhance its robustness and clinical utility.

## Figures and Tables

**Figure 1 biomimetics-10-00642-f001:**
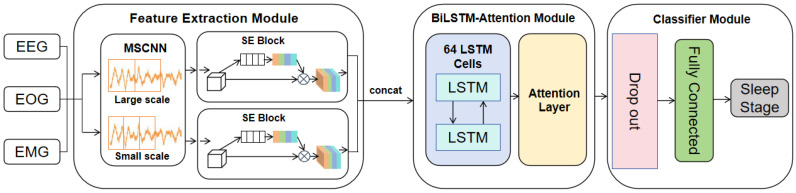
MASleepNet Model.

**Figure 2 biomimetics-10-00642-f002:**
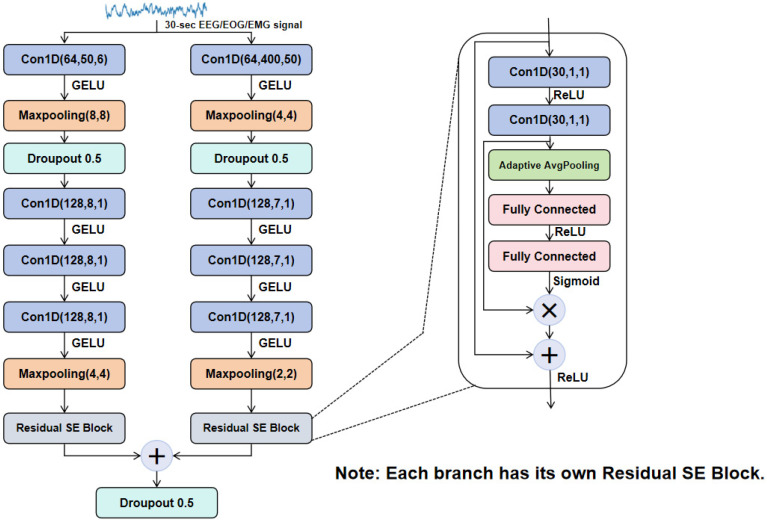
Multi-modal Scalable Feature Extraction Module.

**Figure 3 biomimetics-10-00642-f003:**
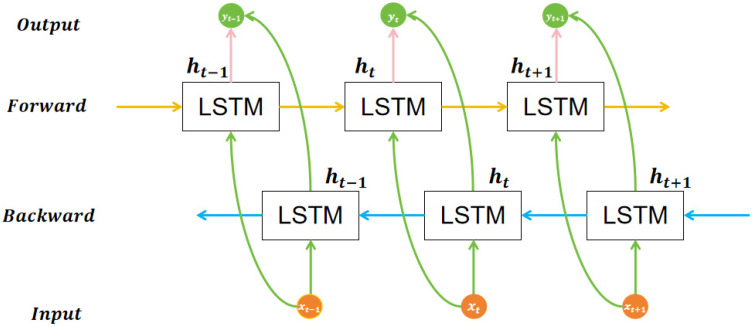
Bi-LSTM Network.

**Figure 4 biomimetics-10-00642-f004:**
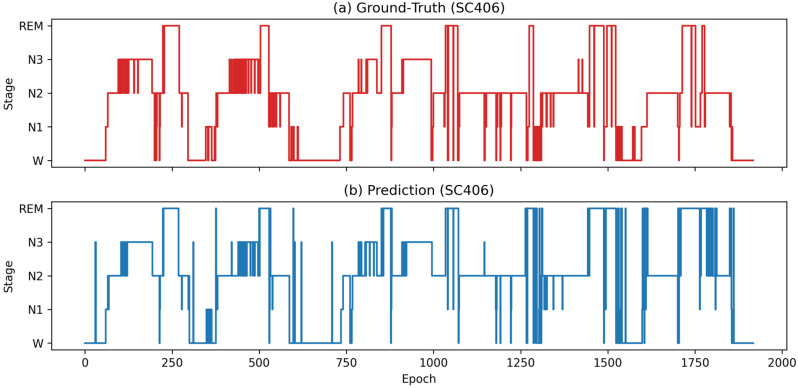
Visualization of the estimated hypnogram for subject SC406 of the: SleepEDF-78 dataset: (**a**) ground-truth hypnogram and (**b**) output hypnogram.

**Figure 5 biomimetics-10-00642-f005:**
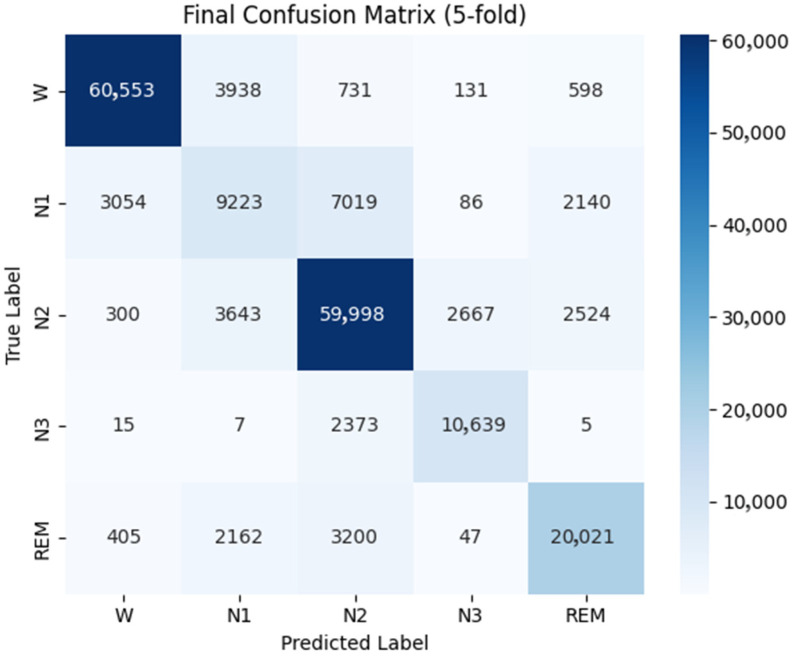
Confusion matrix on the Sleep-EDF-78 dataset.

**Figure 6 biomimetics-10-00642-f006:**
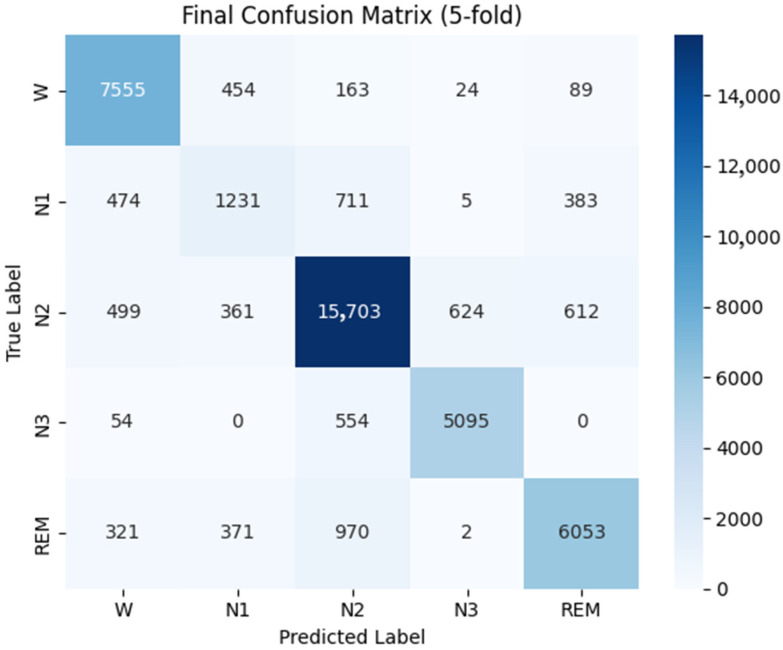
Confusion matrix on the Sleep-EDF-20 dataset.

**Table 1 biomimetics-10-00642-t001:** Sleep-EDF dataset.

Dataset	W	N1	N2	N3	REM	TOTAL
Sleep-EDF-20	8285	2804	17,799	5703	7717	42,308
	19.6%	6.6%	42.1%	13.5%	18.2%	
Sleep-EDF-78	65,951	21,522	69,132	13,039	25,835	195,479
	14.3%	3.2%	43.7%	18.5%	20.3%	

**Table 2 biomimetics-10-00642-t002:** Overall experimental results on the Sleep-EDF-20 dataset and Per-class precision results on the Sleep-EDF-20 dataset.

Model	Overall Results (%)						
	ACC	F1_Macro_	Kappa	Sen	Spe		Pre
DeepSleepNet [23]	80.66	76.60	74.14	80.07	95.21		75.75
ResNetLSTM [36]	74.58	69.97	65.96	71.97	93.54		71.14
AttnSleep [24]	81.39	77.07	74.98	79.71	95.32		77.02
TinySleepNet [37]	80.50	71.40	73.16	72.09	94.81		72.62
SeqSleepNet [15]	75.08	61.68	65.00	63.00	93.03		63.10
MASleepNet	84.53	78.88	78.53	78.43	95.69		80.21
**Model**	**Pre-Class Precisions (%)**						
	**Pre (N1)**	**Pre (N2)**	**Pre (N3)**	**Pre (REM)**		**Pre (Wake)**	
DeepSleepNet [23]	36.85	91.64	79.08	80.12		89.58	
ResNetLSTM [36]	28.29	85.91	78.53	74.83		82.38	
AttnSleep [24]	37.78	90.51	82.04	83.84		88.45	
TinySleepNet [37]	36.45	87.69	83.12	70.67		82.05	
SeqSleepNet [15]	24.09	81.02	82.56	59.32		78.66	
MASleepNet	53.55	86.30	88.04	84.06		87.29	

**Table 3 biomimetics-10-00642-t003:** Overall experimental results on the Sleep-EDF-78 dataset and Per-class precision results on the Sleep-EDF-78 dataset.

Model	Results (%)						
	ACC	F1_Macro_	Kappa	Sen	Spe		Pre
DeepSleepNet [23]	76.87	72.25	69.06	76.64	94.36		70.86
ResNetLSTM [36]	76.77	72.89	68.90	76.43	94.31		71.75
AttnSleep [24]	75.70	71.72	67.54	75.31	94.07		70.70
TinySleepNet [37]	78.06	69.39	69.29	69.31	94.07		71.01
SeqSleepNet [15]	73.84	62.02	63.27	63.52	92.92		64.75
MASleepNet	82.56	76.12	74.95	75.85	95.16		76.94
**Model**	**Pre-Class Precisions (%)**						
	**Pre (N1)**	**Pre (N2)**	**Pre (N3)**	**Pre (REM)**		**Pre (Wake)**	
DeepSleepNet [23]	39.48	85.27	57.82	75.05		96.27	
ResNetLSTM [36]	37.63	86.11	65.57	74.29		94.65	
AttnSleep [24]	35.79	85.08	64.03	71.04		96.48	
TinySleepNet [37]	40.16	78.96	80.21	65.33		89.06	
SeqSleepNet [15]	32.44	79.88	70.50	55.47		80.66	
MASleepNet	47.64	81.23	77.96	79.97		92.78	

**Table 4 biomimetics-10-00642-t004:** Comparative experimental results on the Sleep-EDF-20 dataset.

Model	Results (%)					
	ACC	F1_Macro_	Kappa	Sen	Spe	Pre
No-MSConv	82.07	75.66	75.23	75.57	95.09	76.71
EEG-Only	82.73	74.90	76.17	75.01	95.32	77.31
No-SE	84.22	78.33	78.34	78.14	95.62	79.06
No-LSTM	84.18	78.34	78.12	78.18	95.65	79.56
No-Attn	84.13	78.22	78.06	78.08	95.63	78.95
MASleepNet	84.53	78.88	78.53	78.43	95.69	80.21

**Table 5 biomimetics-10-00642-t005:** Comparative experimental results on the Sleep-EDF-78 dataset.

Dataset	Results (%)					
	ACC	F1_Macro_	Kappa	Sen	Spe	Pre
No-MSConv	79.77	72.69	71.76	72.86	94.49	73.80
EEG-Only	79.94	72.52	72.07	72.41	94.65	73.57
No-SE	82.05	75.95	74.61	75.36	95.08	76.62
No-LSTM	81.52	73.56	74.29	72.34	94.90	76.51
No-Attn	81.85	75.88	74.78	75.68	95.14	76.48
MASleepNet	82.56	76.12	74.95	75.85	95.16	76.94

## Data Availability

Restrictions apply to the availability of these data. Data were obtained from [physiobank] and are available from the authors/at Sleep-EDF Database Expanded.

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
