# Peer review of "MASleepNet: A Sleep Staging Model Integrating Multi-Scale Convolution and Attention Mechanisms"

_biomimetics, 2025, doi:10.3390/biomimetics10100642_

Round 1
Reviewer 1 Report
Comments and Suggestions for Authors
This paper presents MASleepNet, a multimodal deep learning model for automatic sleep staging. While the work contributes valuable insights to sleep staging research, several critical areas require refinement to enhance its scientific rigor, clarity, and practical utility, as detailed below:
(1) in abstract, several abbreviations are not defined, such as PSG, BiLSTM. It is better to give their full name for easy to understand.
(2) The paper’s figures fall short of effectively supporting key technical claims and experimental findings, undermining reader comprehension. For example, an arrow is missed in figure 1from LSTM to Attention Layer. Dropout should be vertically arranged. In figure 2, 30-sec single may be signal. The MFEM module in figure 2 differs from that given in figure 1. Figure 2 uses one SE block in MFEM, while it has two SE block in figure 1. The authors have to check this seriously.
(3) figure 4 is explained in line 344; while figure 5 is explained in line 326. It is better to explain the figures in order. Section 3.3 and section 3.5 have the same title. Table 2 and table 3 are not well-organized. It is better to split them to four tables for clear presentation.
(4) the ablation experiment is not enough to analyze the method. For example, in lines 103-122, the authors say three contributions; hence, it is better to add experiment about without SE module and without BiLSTM module.
(5) some grammars. In lines 141-142, EOG, and EMG is repeated. In addition, four channels mean the four EEG channels, but how about the channel numbers of EOG and EMG. In line 183, X_j, W_ij, b_j are wrong equation symbols. The authors have to check all formula in the paper. In lines 155 and 157, MFEM is repeatedly defined. In line 501-502, references 4 and 5 is wrongly given. Moreover, the state-of-the-art references is not reviewed. It is better to acknowledge some related papers in the past two years.
Author Response
Comments 1: in abstract, several abbreviations are not defined, such as PSG, BiLSTM. It is better to give their full name for easy to understand.
Responses 1: We appreciate the reviewer’s suggestion. We have revised the abstract to provide the full names of the abbreviations at their first occurrence. For example, “Polysomnography (PSG)” and “Bidirectional Long Short-Term Memory (BiLSTM)” are now explicitly stated. The corresponding modifications have been highlighted in red in the revised manuscript.
Comments 2: The paper’s figures fall short of effectively supporting key technical claims and experimental findings, undermining reader comprehension. For example, an arrow is missed in figure 1from LSTM to Attention Layer. Dropout should be vertically arranged. In figure 2, 30-sec single may be signal. The MFEM module in figure 2 differs from that given in figure 1. Figure 2 uses one SE block in MFEM, while it has two SE block in figure 1. The authors have to check this seriously.
Responses 2: We sincerely thank the reviewer for this valuable comment, which has greatly helped us improve the quality and clarity of the figures. We have carefully revised the manuscript as follows:
In Figure 1, the missing arrow from LSTM to Attention Layer has been added, and the Dropout module has been vertically aligned for clarity.
In Figure 2, “30-sec single” has been corrected to “30-sec EEG/EOG/EMG signal”.
Regarding the MFEM module, we have revised Figure 2 to ensure consistency with Figure 1. Specifically, the MFEM module in Figure 2 has been updated to include two SE blocks, making it consistent with the architecture described in Figure 1.
Comments 3: figure 4 is explained in line 344; while figure 5 is explained in line 326. It is better to explain the figures in order. Section 3.3 and section 3.5 have the same title. Table 2 and table 3 are not well-organized. It is better to split them to four tables for clear presentation.
Responses 3: We sincerely appreciate the reviewer’s constructive feedback, which has helped us improve the overall clarity and organization of the manuscript. We have carefully revised the paper as follows:
The explanations of Figure 4 and Figure 5 have been rearranged so that the figures are explained sequentially in the correct order (Line 318–343).
The duplicate titles of Section 3.3 and Section 3.5 have been corrected to ensure clear distinction.
Table 2 and Table 3 have been reorganized and split into four separate tables for better readability and clearer presentation of results.
Comments 4: the ablation experiment is not enough to analyze the method. For example, in lines 103-122, the authors say three contributions; hence, it is better to add experiment about without SE module and without BiLSTM module.
Responses 4: We sincerely thank the reviewer for this valuable suggestion. Following your advice, we have conducted additional ablation experiments to further analyze the contributions of each component in MASleepNet:
We implemented a version without the SE module.
We implemented a version without the BiLSTM module.
The corresponding results have been added to the revised manuscript (new Table 4 and Table 5). These additional experiments further confirm that the SE module effectively enhances discriminative feature channels, while the BiLSTM module is crucial for modeling long-term temporal dependencies (Line 432–438).
Comments 5: some grammars. In lines 141-142, EOG, and EMG is repeated. In addition, four channels mean the four EEG channels, but how about the channel numbers of EOG and EMG. In line 183, X_j, W_ij, b_j are wrong equation symbols. The authors have to check all formula in the paper. In lines 155 and 157, MFEM is repeatedly defined. In line 501-502, references 4 and 5 is wrongly given. Moreover, the state-of-the-art references is not reviewed. It is better to acknowledge some related papers in the past two years.
Responses 5:We sincerely thank the reviewer for pointing out these important issues. We have carefully revised the manuscript as follows:
Lines 142–144: The repeated mention of EOG and EMG has been removed. The channel description has been clarified as two EEG channels (Fpz–Cz, Pz–Oz), one EOG channel, and one EMG channel, to avoid misunderstanding.
Line 186-188: The equation symbols X_j, W_ij, b_j were incorrect. We have corrected them and carefully re-checked all mathematical formulas throughout the manuscript to ensure consistency and correctness.
Lines 157 and 158: The redundant definition of MFEM has been removed. Now, the module is introduced only once in a concise and coherent way.
Lines 512–514: The mis-citation of References 4 and 5 has been corrected.
References: Following the reviewer’s suggestion, we have updated and expanded the reference list to include more than 30 works, incorporating recent state-of-the-art studies from the past two years (e.g., Lines 494-497, 529-532). These additions strengthen the related work section and provide a more comprehensive overview of the field.
Summary for comments:
We sincerely thank the reviewer for the constructive feedback. According to the suggestions, we have thoroughly revised and polished the manuscript. Specifically, the figures and tables have been carefully re-drawn and improved for better clarity and consistency; the textual descriptions corresponding to each figure/table have been reorganized to ensure logical order and readability; and all modifications have been highlighted in red in the revised manuscript.

Reviewer 2 Report
Comments and Suggestions for Authors
The work structured well and including all essential parts for presenting the model, dataset and results.
1- My only comment is adding clear explanation behind defining model hyperparameters.
Author Response
Comments 1: My only comment is adding clear explanation behind defining model hyperparameters.
Responses 1:
We sincerely thank the reviewer for this valuable suggestion. Following the advice, we have added a detailed explanation of the rationale behind selecting model hyperparameters, including convolution kernel sizes, hidden dimensions, learning rate, batch size, and dropout ratio. These explanations are now provided in the revised manuscript (Lines 286–296, highlighted in red). The revisions clarify how each hyperparameter was determined based on prior literature, empirical validation, and task-specific requirements, thereby improving the transparency and reproducibility of the proposed model.
Summary:
We appreciate the reviewer’s insightful suggestion. The manuscript has been revised accordingly, with clear justifications for model hyperparameter settings, ensuring that the methodology is better explained and more comprehensible for readers.

Reviewer 3 Report
Comments and Suggestions for Authors
Paper title: MASleepNet: A Sleep Staging Model Integrating Multi-Scale Convolution and Attention Mechanisms
This paper studies MASleepNet, a sleep staging neural network model that integrates multimodal deep features.This model takes multi-channel PSG signals (including EEG (Fpz-Cz, Pz-Oz), EOG, and EMG) as input and employs a multi-scale convolutional module to extract features at different time scales in parallel. It then adaptively weights and fuses the features from each modality using a channel-wise attention mechanism. The fused temporal features are fed into a BiLSTM sequence encoder, where an attention mechanism is introduced to identify key temporal segments. Finally, a fully connected layer outputs the sleep stage classification result. The model was experimented on the publicly available Sleep-EDF dataset (consisting of two subsets, Sleep-EDF-78 and Sleep-EDF-20). The proposed model achieved an accuracy of 82.56% and 84.53% on the Sleep-EDF-78 and Sleep-EDF-20 datasets, respectively. Authors show some comparison tables with previous approaches with some advantages of MASleepNet.
The approach shows the advantages compared to another paper, which was published from 2024 to 2025. I have some questions:
- Figures 5-6 have a bad quality when we print them out. Authors should save all figures with .jpg, .emf, or .bitmap.
- Figure 3 is just shown in the manuscript, but I did not see any explain about it. How to apply Bi-LSTM for your approach.
- Explain more about Figures 5-6. It is the performance of SleepEDF-78 and SleepEDF- 325 datasets.
- Section 5 should be separated into 2 sections: future direction and conclusion.
- References just have to be before 2023. Authors should cite some papers from 2025 and 2024.
- I check similarity. It is around 26%. Authors should revise it.
In my point of view, this paper should have minor revision before publishing.
Author Response
Comments 1: Figures 5-6 have a bad quality when we print them out. Authors should save all figures with .jpg, .emf, or .bitmap.
Responses 1: We sincerely thank the reviewer for pointing out this issue. Following the suggestion, we have redrawn Figures 5 and 6 with higher resolution and saved them in .jpg format to ensure clarity and readability when printed. The updated figures have been incorporated into the revised manuscript (see Figures 5 and 6, Lines 314-317, highlighted in red).
Comments 2: Figure 3 is just shown in the manuscript, but I did not see any explain about it. How to apply Bi-LSTM for your approach.
Responses 2: We sincerely thank the reviewer for this valuable suggestion. In the revised manuscript, we have added a detailed explanation of how the Bi-LSTM is applied in our approach. Specifically, the Bi-LSTM network is employed to capture contextual dependencies in the feature sequence obtained after multimodal feature fusion. As shown in Figure 3, the input sequence is processed in both forward and backward directions, and the hidden states from both directions are concatenated to form the bidirectional representation (see Lines 224-230, highlighted in red). Furthermore, the Bi-LSTM output is subsequently fed into the attention layer to focus on the most informative temporal segments for classification.
Comments 3: Explain more about Figures 5-6. It is the performance of SleepEDF-78 and SleepEDF- 20 datasets.
Responses 3: We greatly appreciate the reviewer’s constructive suggestion. In the revised manuscript, we have expanded the explanation of Figures 5 and 6 to provide a clearer interpretation of the model’s performance on the SleepEDF-78 and SleepEDF-20 datasets. Specifically, we emphasized that although the model achieves high accuracy in major sleep stages such as N2, N3, REM, and W, the N1 stage remains the most challenging due to its short duration and smooth transitions with adjacent stages. Furthermore, we highlighted that despite the differences between the two datasets in terms of sample size, signal type, and subject distribution, the model still demonstrates stable generalization ability across both datasets. The detailed explanation has been added in Section 3.4 (Lines 329-343, highlighted in red).
Comments 4: Section 5 should be separated into 2 sections: future direction and conclusion.
Responses 4: We sincerely thank the reviewer for this helpful suggestion. In the revised manuscript, Section 5 has been reorganized into two separate parts: Conclusion and Future Directions. The Conclusion section now summarizes the main contributions and experimental findings of MASleepNet, while the Future Directions section outlines potential improvements, including model lightweight design, real-time deployment in portable devices, and validation on real-world clinical datasets. This adjustment improves the logical clarity and readability of the manuscript (Lines 447-479, highlighted in red).
Comments 5: References just have to be before 2023. Authors should cite some papers from 2025 and 2024.
Responses 5: We sincerely thank the reviewer for pointing this out. In the revised manuscript, the reference list has been updated by including several recent works published in 2024 and 2025 that are closely related to multimodal deep learning, sleep staging, and attention-based neural networks. These additional references provide stronger support for the state-of-the-art background and ensure that the literature review remains up-to-date (Lines 494-497, 529-532, highlighted in red).
Comments 6: I check similarity. It is around 26%. Authors should revise it.
Responses 6: We sincerely thank the reviewer for raising this important issue. The manuscript has been carefully revised to reduce textual overlap and improve originality.

Round 2
Reviewer 1 Report
Comments and Suggestions for Authors
The authors have done a good job revising their paper, producing a sound paper that is worthy of publication.